# Atmospheric lifetime of sulphur hexafluoride ($SF_6$) and five other trace gases in the BASCOE model driven by three reanalyses

Sarah Vervalcke[1], Quentin Errera[1], Roland Eichinger[2], Thomas Reddmann[3], Simon Chabrillat[1], Marc Op de beeck[1], Gabriele Stiller[3], and Emmanuel Mahieu[4]

[1]Royal Belgian Institute for Space Aeronomy (BIRA-IASB), Brussels, Belgium
[2]Institut für Physik der Atmosphäre, Deutsches Zentrum für Luft- und Raumfahrt (DLR), Oberpfaffenhofen, Germany
[3]Institute for Meteorology and Climate Research, Karlsruhe Institute of Technology (KIT), Karlsruhe, Germany
[4] Institute of Astrophysics and Geophysics, Université de Liège (ULg), Liège, Belgium

**Correspondence:** Sarah Vervalcke (sarah.vervalcke@aeronomie.be) and Quentin Errera (quentin.errera@aeronomie.be)

## Abstract

In this work, sulphur hexafluoride ($SF_6$), which is often used as a tracer for stratospheric transport due to its inertness in the stratosphere and nearly linear growth rate in the troposphere, is included in the chemistry transport model (CTM) of the Belgian Assimilation System for Chemical ObsErvations (BASCOE). Sink and recovery reactions for this species are implemented in the model, which has a top in the mesosphere at 0.01 hPa. The simulated $SF_6$ distributions are compared with MIPAS and ACE-FTS observations and the global atmospheric lifetime is computed from CTM runs driven by three recent meteorological reanalyses: ERA5, MERRA2 and JRA-3Q. The results show that BASCOE $SF_6$ profiles are generally within 10% of the satellite observations below 10 hPa, although discrepancies increase at higher altitudes. The global atmospheric lifetime is used as an additional diagnostic for the implementation of the chemistry in the mesosphere, where satellite measurements are unavailable. The derived $SF_6$ lifetimes are 2646 years with ERA5, 1909 years with MERRA2 and 2147 years with JRA-3Q, in accordance with recent literature. Due to the large spread of published lifetimes for $SF_6$, the study is extended to $N_2O$, $CH_4$, CFC-11, CFC-12 and HCFC-22, to validate the $SF_6$ results. The lifetimes for these species are in agreement with previously reported values, and their spread between simulations is smaller compared to $SF_6$. This analysis highlights the sensitivity of $SF_6$ to the input reanalysis data sets and thus to differences in dynamics.

## 1   Introduction

Studies of middle atmospheric transport have been important for a long time and are motivated in part by the existence of the stratospheric ozone layer that protects the Earth from solar UV radiation. Transport of trace gases in the stratosphere is dominated by a large scale circulation pattern that is part of the Brewer-Dobson circulation (BDC), discovered by A. Brewer and G. Dobson (Brewer, 1949; Dobson, 1956; Dobson et al., 1930). The BDC is the result of an uplift of air from the tropical tropopause into the stratosphere and a consequent poleward transport driven by planetary wave breaking in the atmosphere. At mid- and high latitudes the air returns to the troposphere upon which it can be recirculated. The BDC is important for the transport of trace gases such as ozone, influencing both the chemistry and climate of the atmosphere. A comprehensive review of the Brewer-Dobson circulation can be found in Butchart (2014).

Climate models predict an increase in the strength of the BDC (Butchart et al., 2006; Hardiman et al., 2013; Abalos et al., 2021), typically quantified by the tropical mass upwelling. While strength and speed describe different aspects of the circulation, they tend to vary together, as shown in Austin and Li (2006). A common way to diagnose the speed of the BDC is through age of air (AoA) studies. These studies look at the (average) time an air parcel takes to move from a reference surface in the troposphere to the stratosphere. Changes in AoA at a given location in the stratosphere thus reflect changes in circulation speed: the transportation time increases when the circulation slows down and decreases when the circulation speeds up. Additionally, two-way mixing between the tropics and the extra-tropics can increase stratospheric AoA by making air parcels recirculate, while in the extra-tropical lower stratosphere, mixing rejuvenates the air. Mixing within the extra-tropical stratosphere also has local effects and flattens the latitudinal age gradient (Garny et al., 2014; Dietmüller et al., 2018; Eichinger et al., 2019). The concept of AoA is described in Hall and Plumb (1994) and Garny et al. (2024b). The AoA can be computed from a synthetic, idealized model tracer or from real long-lived trace gases with a nearly linear increase in the troposphere, such as carbon dioxide ($CO_2$) and sulphur hexafluoride ($SF_6$). $SF_6$ in particular is often used because of its inertness in the stratosphere and the absence of strong seasonal variations. The emissions of $SF_6$ at the surface are almost completely anthropogenic due to its use in high voltage insulation for the transmission and distribution of electricity. Additionally, $SF_6$ is a potent greenhouse gas with a global warming potential that is estimated to be 22,500 times larger than that of $CO_2$ (Wang et al., 2019) and is therefore important for climate studies. However, $SF_6$ suffers from a sink reaction in the mesosphere which makes the age of air older than what is expected from an inert tracer. Moreover, $SF_6$ is a noisy data product because its spectral signatures are close to those of $CO_2$ and $H_2O$, and due to its low abundance in the atmosphere, on the order of parts per trillion, the signal is weak. While this provides challenges when using $SF_6$ as a tracer of the residual circulation, Garny et al. (2024a) propose a correction for the ageing due to the mesospheric loss and Voet et al. (2025) combines several long-lived trace gases to compute the AoA, which reduces the uncertainty compared to a method based on $SF_6$ alone.

While AoA trends are a useful diagnostic, discrepancies exist between AoA trends computed from observations, models and reanalyses (Garny et al., 2024b). In Chabrillat et al. (2018), the difference in absolute AoA between the CTM driven by different reanalyses is at most 1 year in the tropical lower stratosphere and up to 2 years or more in the upper stratosphere, with trends over 2002-2015 ranging from -0.4 to 0.4 yr dec$^{-1}$ above 10 hPa. Chemistry climate models suggest decreasing stratospheric age of air trends around 0.1-0.2 yr dec$^{-1}$. Balloon observations, on the other hand, show positive trends on the order of 0.1-0.3 yr dec$^{-1}$ (Garny et al., 2024b). The cause of these discrepancies is not clear.

In this work, the implementation of $SF_6$ in the Belgian Assimilation System for Chemical ObsErvations (BASCOE) chemistry transport model (CTM) is described. The implementation of $SF_6$ is evaluated using independent satellite observations from MIPAS (Michelson Interferometer for Passive Atmospheric Sounding, Fischer et al., 2008) and ACE-FTS (Atmospheric Chemistry Experiment Fourier Transform Spectrometer, Bernath et al., 2005), and via the computation of the global atmospheric lifetime of $SF_6$, which is compared with the literature. Three recent meteorological reanalyses have been used to drive the BASCOE simulations: ERA5 (ECMWF Reanalysis V5, Hersbach et al., 2020), MERRA2 (Modern-Era Retrospective analysis for Research and Applications V2, Gelaro et al., 2017) and JRA-3Q (Japanese Reanalysis for Three Quarters of a Century, Kosaka et al., 2024). These reanalyses extend upwards to 0.01 hPa in the mesosphere where $SF_6$ chemistry is important. In

this study we use three different sets of reanalysis data to drive the BASCOE model in order to analyse the influence of the
meteorology on the results. Due to a large spread of the lifetime values, both those presented here as well as those found in the
literature, the work was extended to five other long-lived species to support the evaluation of our methods and results: $N_2O$,
$CH_4$, CFC-11, CFC-12 and HCFC-22. These six species can be used to compute the AoA using the method from Voet et al.
(2025). However, AoA diagnostics are not presented in this work.

The paper is organised as follows: Section 2 introduces the different data sets used. This includes MIPAS and ACE-FTS
observations and the three reanalyses. Section 3 discusses the BASCOE model simulations, and in particular the implemen-
tation of the chemistry of $SF_6$ and the set-up of the model. Section 4 describes the computation of the global atmospheric
lifetime. Finally, the comparison between the model simulations and the observations is presented in Sect. 5, along with the
obtained values for the global lifetime. The conclusions are presented in Sect. 6. While this study compares diagnostics of the
reanalyses, it does not present a full inter-comparison of the reanalyses themselves.

## 2 Data sets

The following sections describe the data sets that are used in this work. The first two subsections deal with the satellite data
that are used to evaluate the model results. The last subsection informs about the reanalysis data sets that are used as input
for the BASCOE CTM. We used two satellite data sets as observational reference, namely MIPAS and ACE-FTS. These are
described in more detail below. As drivers for BASCOE, three different reanalyses were used, namely ERA5, MERRA2, and
JRA-3Q. Details about these reanalyses are given below.

### 2.1 MIPAS

The Michelson Interferometer for Passive Atmospheric Sounding (MIPAS, Fischer et al., 2008), onboard the ENVISAT satel-
lite, operated from July 2002 until April 2012 in a sun-synchronous low-Earth orbit. It is a Fourier Transform Spectrometer
with a limb viewing geometry, measuring atmospheric infrared radiation from which vertical profiles of trace gases are re-
trieved. MIPAS was designed to measure over 20 species, including $SF_6$, CFC-11 and CFC-12, $CH_4$, $N_2O$ and HCFC-22. In
April 2004, MIPAS had a failure and observations were halted until January 2005, after which measurements resumed with
reduced spectral resolution. MIPAS datasets are thus typically split into two phases: one with high spectral resolution and one
with reduced spectral resolution. In this study, the model output is evaluated using MIPAS V8 data from the IMK/IAA research
processor, developed by the Institut für Meteorologie und Klimaforschung (IMK) and the Instituto de Astrofísica de Andalucía
(IAA). To judge the model differences with respect to MIPAS and ACE-FTS observations, we compare with typical instrumen-
tal differences from satellite validation studies. These validation studies are summarized in Table 1 with their data versions.
Most of the instrument differences between ACE-FTS and MIPAS are between 10 and 20%, particularly for the species $CH_4$,
$N_2O$ and CFC-12. However, the agreement depends on the altitude and differences increase significantly above 50 hPa for
some species.

**Table 1.** Summary of observational differences between ACE-FTS and MIPAS from validation studies. The data versions used in this work are MIPAS IMK/IAA V8 and ACE-FTS V5.3.

| Species | Validation study | Data versions | Agreement |
|---|---|---|---|
| $SF_6$ | SPARC (2017). | ACE-FTS V2.2, MIPAS spectral version 5 | $\pm2.5\%$ difference with multi-instrument mean below 50 hPa for annual mean profiles. Above 50 hPa, mostly within $\pm5\%$, some regions $\pm10\%$ to $\pm20\%$. Monthly mean profiles generally consistent, with slightly larger deviations for some months. |
| $N_2O$ | Sheese et al. (2017) | ACE-FTS V3.5, MIPAS spectral version 5, MLS V3.3 | ACE-FTS shows -9% to +7% differences with MIPAS and -21% to 5% with MLS below 35km. Above 35 km, agreement within $\pm 10\%$. |
| CFC-11 | SPARC (2017) | ACE-FTS V2.2, MIPAS spectral version 5 | MIPAS monthly mean profiles show 5-10% difference with ACE-FTS below 100 hPa. ACE-FTS shows up to 50% difference to multi-instrument mean at 30 hPa. |
| CFC-12 | SPARC (2017) | ACE-FTS V2.2, MIPAS spectral version 5 | Differences between monthly mean profiles within $\pm10\%$ across the vertical range. |
| $CH_4$ | Laeng et al. (2015) | ACE-FTS V3.5, MIPAS spectral version 5 | 12% difference between 20 and 65 km. Difference of 15% at 17 km. |
| | SPARC (2017) | ACE-FTS V2.2, HALOE V19, MIPAS spectral version 5 | Differences of between the monthly mean profiles of up to $\pm20\%$ to the multi-instrument mean. |
| HCFC-22 | Kolonjari et al. (2024) | ACE-FTS V5.2, MIPAS IMK/IAA V8 | +3 to 10 % difference between 5 and 10km. $\pm5$ % difference between 10 and 21 km. +5 to 14 % difference between 21 and 25 km. |

## 2.2 ACE-FTS

The Atmospheric Chemistry Experiment Fourier Transform Spectrometer (ACE-FTS) is a Canadian-led mission on SCISAT-1 (Bernath et al., 2005). It started taking measurements in February 2004 and is still operational after more than 20 years on orbit.

This instrument performs infrared solar-occultation measurements on an inclined circular orbit. ACE-FTS takes measurements twice per orbit, during sunrise and sunset, allowing for up to 30 observations per day. ACE-FTS focuses on the region in the stratosphere that contains the ozone layer and measures 70 atmospheric trace gases, including CFC-11, CFC-12, $CH_4$, $N_2O$ and HCFC-22. The data version used in this work is V5.3.

## 2.3 Reanalyses

Three different reanalysis data sets are used to drive the BASCOE chemistry transport model. Reanalyses offer a globally complete data set of atmospheric variables by assimilating observational data into models to ensure spatial and temporal homogeneity. Reanalyses with a top around 0.01 hPa were chosen to capture mesospheric circulation features that are important for the chemistry of $SF_6$.

The first reanalysis is the ERA5 (Hersbach et al., 2020) data set, issued by the European Centre for Medium-Range Weather Forecasts (ECMWF). It is available from 1940 onwards. The ERA5 atmospheric data has a horizontal resolution of 31km ($\sim 0.28125°$), and 137 vertical levels from the surface up to 0.01hPa.

Similarly, the Modern-Era Retrospective analysis for Research and Applications version 2, or (MERRA2, Gelaro et al., 2017), is a data set from the Global Modelling and Assimilation Office (GMAO) which covers the period 1980 to present. The data set has a horizontal resolution of $0.5°\mathrm{x}0.625°$ and 72 vertical levels, from the surface up to 0.015 hPa. MERRA2 is the only one of the three reanalysis data sets used here that assimilates Aura Microwave Limb Sounder (Aura MLS) temperature profiles from 2004 onwards above 5 hPa (SPARC, 2022).

Finally, the Japanese Reanalysis for Three Quarters of a Century (JRA-3Q, Kosaka et al., 2024) is the third long-term reanalysis produced by the Japan Meteorological Agency (JMA), covering the period from September 1947 to present. This is the most recently released data set of the three reanalyses used in this work. JRA-3Q has a higher resolution than its predecessor JRA-55: it has a horizontal resolution of 40km ($\sim 0.359°$) and 100 vertical levels from the surface to 0.01hPa.

Each reanalysis has a sponge layer at the model top to avoid unphysical reflection of wave energy at a rigid model lid. These sponge layers are implemented differently in each reanalysis and can explain some of the differences found at the upper levels and in our results. According to SPARC (2022, Chap. 5) the climatological wave driving and the climatological circulation strength and structure agree closely among the most recent reanalysis products available at the time (MERRA2, ERA5 and JRA-55), however, MERRA2 is shown to have slower upwelling in the tropics than the other reanalyses. This slower BDC is confirmed by tracer-transport studies, indicating that the difference is not only related to the radiation budget, but also to transport (SPARC, 2022, Chap. 5). The SPARC report advises to use MERRA2 with caution in Brewer-Dobson circulation studies and for many BDC diagnostics, JRA-55 and ERA-Interim are more suitable (with limitations). Ploeger et al. (2021) showed that ERA5 has a slow circulation compared to ERA-Interim and found similarities between ERA5 and MERRA2 in the tropical upwelling and the evolution of the residual circulation transit times. ERA-Interim and JRA-55 are not used here because these do not reach high enough to capture the mesospheric chemistry of $SF_6$.

## 3 The BASCOE model

The model that is used in this work is the Chemistry Transport Model (CTM) of the Belgian Assimilation System for Chemical ObsErvations (BASCOE, Errera et al., 2008). This CTM is dedicated to stratospheric composition and includes around 60 chemical species. All species are advected by the flux-form semi-Lagrangian scheme (Lin and Rood, 1996) which conserves mass and preserves tracer-tracer correlations. Approximately 200 chemical reactions (gas phase, photolysis and heterogeneous) are taken into account, including a parametrization of Polar Stratospheric Cloud (PSC) microphysics (Errera et al., 2019). The gas-phase and photolysis reaction rates have been updated according to Burkholder et al. (2015). Chemical reactions are solved by a chemical kinetic preprocessor (KPP, Damian et al., 2002). Note that chemistry is not computed in the troposphere at altitudes below 400 hPa. The CTM is driven by meteorological analyses as described in Sect. 3.2. The following subsections first describe the update of the chemistry and then the preprocessing of the dynamical fields and details of the simulations.

### 3.1 Implementation of SF6 chemistry

$SF_6$ is inert in the stratosphere and is destroyed by attachment of electrons in the mesosphere producing a negative excited ion $(SF_6^-)^*$. $SF_6$ is partially recovered via stabilizing reactions, either directly via photodetachment of an electron, or by cycling the stabilized $SF_6^-$ back to $SF_6$ via several other reactions.

In order to model the distribution of $SF_6$, the BASCOE chemistry transport model has been updated with $SF_6$ chemistry. The reaction scheme used in this work was first described in Reddmann et al. (2001) and later used in the EMAC model (Loeffel et al., 2022). It is shown in Table 2. The main destruction mechanism for $SF_6$ is auto-attachment of electrons (R2). UV-photolysis (R1) is not used in BASCOE because the corresponding loss rate is negligible compared to the loss rate from electron auto-attachment (Totterdill et al., 2015). In addition to $SF_6$, the species $SF_6^-$ and $(SF_6^-)^*$ have been added to the list of chemical compounds of BASCOE and the rate constants for the reactions were taken from Reddmann et al. (2001). We only consider the fast electron-attachment reaction and neglect the destructive branch, which accounts for less than 0.1% of the total reaction (Miller et al., 1994). The fast electron-attachment reaction produces an excited negative ion of $SF_6$, which in turn can stabilize and react with H, HCl, O3 or $h\nu$ to form $SF_6$ or other products. In order to implement these chemical reactions, the electron density and the computation of the photodetachment rate are needed. The electron density is parametrized taking as input the altitude, the latitude, the solar zenith angle, the $O_2$ column, the air density and the day of the year, using the same code as in EMAC (Reddmann et al., 2001; Loeffel et al., 2022). The relevant electrons are those from the D-region of the ionosphere (see Brasseur and Solomon (2005) Chap. 7 for a derivation of the electron density), which is located roughly between 60 km and 90-95 km, and thus lies mostly within the mesosphere. The electron density assumes mean solar activity.

We use an electron-attachment rate that is specific for thermalized electrons ($\ll$ 1 eV), resulting in fast attachment. As thermalization is fast at the altitudes relevant for electron-attachment, the use of the full electron density is justified. The photodetachment (R3) rate is calculated within the BASCOE model as a standard photolysis rate, via:

$$j_3(\theta) = \sum_i \sigma_{pd}(\lambda_i) F(\theta, \lambda_i, p) \Delta\lambda_i \tag{1}$$

Where $\sigma_{pd}$ is the cross-section for photodetachment and $F$ is the solar irradiance (actinic flux), dependent on the solar zenith angle $\theta$, the wavelength $\lambda_i$ and the pressure $p$. The cross-sections $\sigma_{pd}$ are computed using a relation proposed by Datskos et al. (1995) for wavelengths below 337 nm and a constant cross-section of $2 \times 10^{-18}$ for wavelengths above 337 nm, as proposed by Ingólfsson et al. (1994). The cross-sections as a function of wavelength and the photodetachment rate for one snapshot of BASCOE output as a function of latitude and corresponding solar zenith angle are shown in Fig. 1.

The loss rates $\alpha$ and the electron density computed in BASCOE were evaluated against published values. Figure 2a shows how the inverse loss rates from Reddmann et al. (2001) compare to the inverse loss rates in BASCOE on January 15, 2002. This plot also contains the loss rates from Totterdill et al. (2015), which agree well with the loss rates in BASCOE. The electron density in BASCOE is compared with other published electron fields in Fig. 2b. This shows that BASCOE follows the ionization profile from Reddmann et al. (2001). BASCOE electron density also agrees well with an observation taken from

Brasseur and Solomon (2005), shown in Fig. 2b.

## 3.2    Model simulations

This work presents three BASCOE CTM simulations of 25 years (1997-2023) driven by ERA5, MERRA2 and JRA-3Q, respectively, with a model time step of 30 minutes (see Table 3 summarizing the BASCOE simulations performed for this study). Reanalysis dynamical fields were reduced in resolution to match the BASCOE spatial grid using a mass-conserving

preprocessor, following the approach in Chabrillat et al. (2018). This was done because simulating chemistry on the native spatial grids of the reanalyses incurs a large computational cost and storage requirements which are not available on the BIRA-IASB computing system. The BASCOE simulations have been carried out with a horizontal resolution of $2°\times2.5°$ latitude-longitude as in Chabrillat et al. (2018), which is sufficient to capture tropical and high-latitude mixing barriers (Strahan and Polansky, 2006). Similarly, the vertical resolution of the reanalyses has been reduced, since chemistry is not computed on most

**Table 2.** Chemical scheme for $SF_6$

| Reaction No. | Reaction | Reaction rate constant | Remarks |
|---|---|---|---|
| R1 | $SF_6 + h\nu \rightarrow$ products | | UV-photolysis |
| R2 | $SF_6 + e^- \rightarrow (SF_6{}^-)^*$ | $k_2 = 270 \times 10^{-9} \mathrm{cm}^{-3} s^{-1}$ | $e^-$ auto-attachment |
| R3 | $SF_6{}^- + h\nu \rightarrow SF_6 + e^-$ | $j_3 \approx 0.02 s^{-1}$ | photodetachment [1] |
| R4 | $SF_6{}^- + H \rightarrow SF_5{}^+ + HF$ | $k_4 = 0.21 \times 10^{-9} \mathrm{cm}^{-3} s^{-1}$ | |
| R5 | $(SF_6{}^-)^* + M \rightarrow SF_6{}^-$ | $k_5 = 0.19 \times 10^{-9} \mathrm{cm}^{-3} s^{-1}$ | stabilization |
| R6 | $(SF_6{}^-)^* \rightarrow SF_6 + e^-$ | $k_6 = 1 \times 10^6 s^{-1}$ | auto-detachment [2] |
| R7 | $SF_6{}^- + HCl \rightarrow$ products | $k_7 = 1.5 \times 10^{-9} \mathrm{cm}^{-3} s^{-1}$ | |
| R8 | $SF_6{}^- + O_3 \rightarrow SF_6 + O_3{}^-$ | $k_8 = 1.2 \times 10^{-9} \mathrm{cm}^{-3} s^{-1}$ | |

[1] The value of $j_3$ is an indication of the final result of the photodetachment calculation.

[2] There is a typo in Table 1 of Reddmann et al. (2001) and the correct value was taken from its Sect. 2.2.

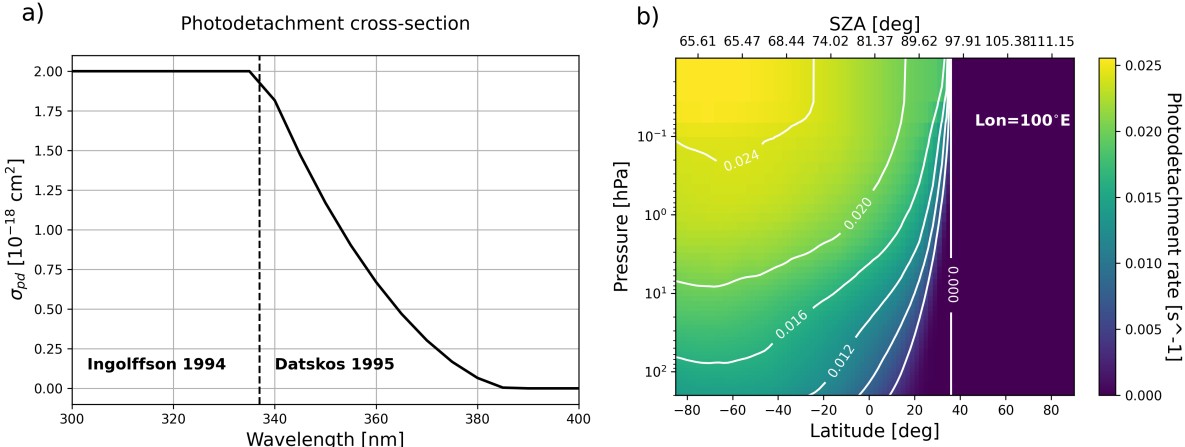

**Figure 1.** Left: photodetachment cross-sections computed from Ingólfsson et al. (1994) and Datskos et al. (1995). Right: photodetachment rate, computed in BASCOE, as a function of latitude with corresponding solar zenith angle (SZA) for a longitude of 100°E on 2002-01-01 at 00:00.

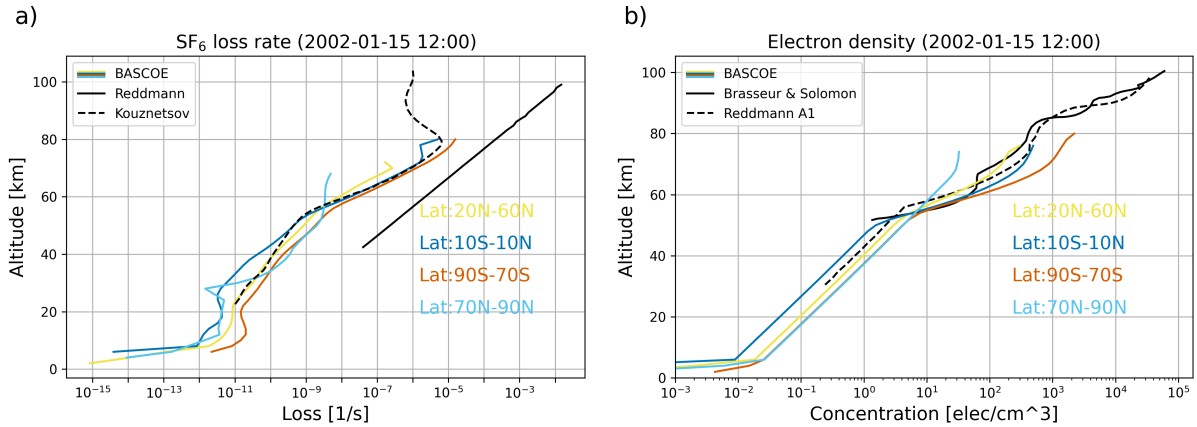

**Figure 2.** Left: comparison between the loss rates as shown in Reddmann et al. 2001 and Totterdill et al. (2015) with a BASCOE snapshot from 2002-01-15 (TRANS_M2_L49) in different latitude bands. Right: example profiles of the electron field of BASCOE, shown over 4 latitude bands, compared with the A and A1 profiles of Reddmann et al. (2001) and the electron density in Brasseur and Solomon (2005), Chap. 7, Fig.7.3.

levels in the troposphere. The vertical regridding procedure is detailed in Appendix A. The native vertical resolution of the reanalyses and those obtained by the vertical regridding are shown in Fig. 3. For ERA5, a grid with 61 levels was chosen, for MERRA2, 49 levels were chosen and finally the grid of JRA-3Q was reduced to 53 levels. From this figure it is clear that the resolution is now lower in the troposphere, thus removing unnecessary levels, while also being somewhat reduced in the stratosphere and mesosphere.

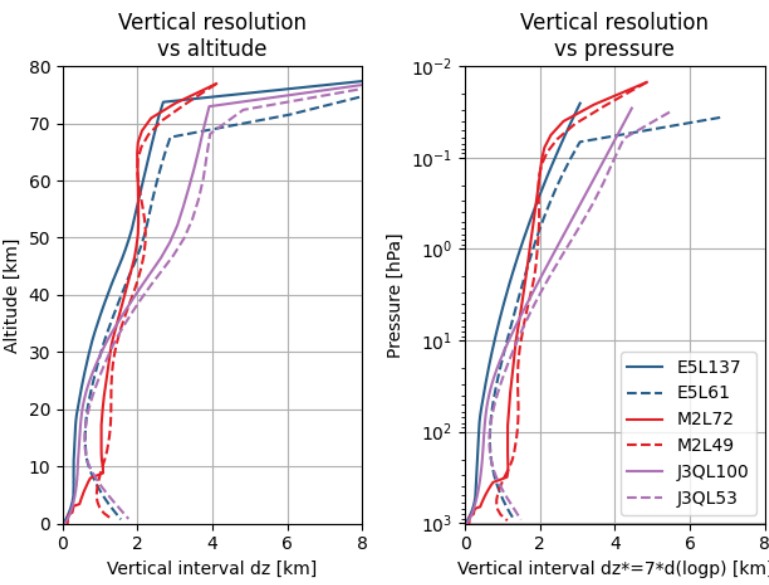

**Figure 3.** Vertical resolution of the different grids as a function of altitude (left) and the pressure (right). Blue, red and purple lines correspond, respectively, to ERA5, MERRA2 and JRA-3Q. Solid and dashed lines correspond, respectively, to native and reduced vertical resolution. Note that the line color of each reanalysis follows the A-RIP conventions, as in other figures.

To assess the transport on the reduced vertical grids, simulations of mean AoA using an idealized clock tracer were performed on both the native and the reduced grids. These simulations re-use meteorological fields from the same two years (1980-1981) of reanalysis data to drive the model in order to avoid the effect of climatological trends. Two years were chosen instead of one to capture the dynamical effects of the Quasi Biennial Oscillation (QBO), which is called a perpetual QBO set-up (Prignon, 2021). Each run lasts 20 years and the results are evaluated at the end of this period to avoid the spin-up of the model.

The average of the model AoA from the last year of the perpetual QBO simulations is shown in Fig. 4. MERRA2 meteorology results in the highest AoA, followed by JRA-3Q, while ERA5 produces the lowest AoA. This could indicate a faster circulation with ERA5 compared to MERRA2 and JRA-3Q in the BASCOE model. The simulations on the reduced and native grids are in good agreement, generally within 0.1 years. Slightly larger differences are found for MERRA2 (0.3 years) in the extra-tropics at 100 hPa, which is acceptable. This validates our choice for the reduced levels.

Initial conditions were based on a BASCOE run driven by ERA5 (see Prignon et al., 2021), except for $SF_6$. For the latter, monthly zonal mean output from an EMAC model simulation was used to initialize the BASCOE simulations on 27-01-1997. This output comes from the specified dynamics simulation described in Loeffel et al. (2022), which has a model top at 0.01 hPa, spans the period 1980-2011 and is nudged to the ERA-Interim reanalysis. The lower boundary conditions for BASCOE were adapted from Meinshausen et al. (2017) for surface emissions from 1700 to 2014, and from Gidden et al. (2019) for
emission projections under the shared socioeconomic pathway SSP2-4.5 beyond 2014.

At every time step of the simulation, BASCOE checks if observations are available for that time and interpolates the model

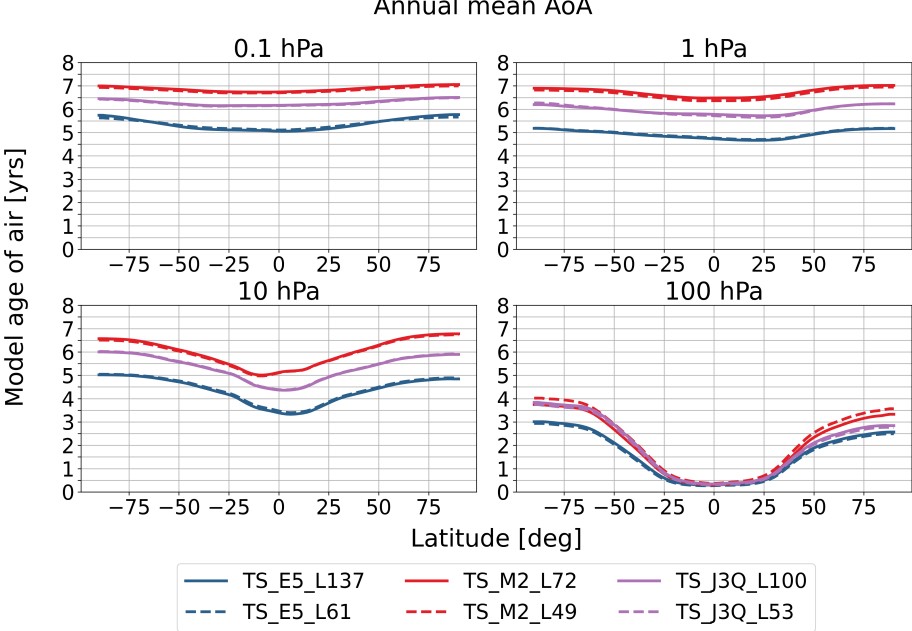

**Figure 4.** Model age of air from perpetual QBO simulations driven by ERA5, MERRA2 or JRA-3Q on their native (solid lines) and reduced (dashed lines) vertical grids, at different altitudes, averaged over the last year of the simulation.

output at the locations of the satellite observations. BASCOE was initially saved on the spatial grid of MIPAS V5 and ACE-FTS V4.1. Although newer data versions were later used, the model output was not re-interpolated to the new data locations, and therefore only subsets of the new data are used. This output in observation space is used to compute statistical differences between the model and the observations, presented in Sect. 5.1.

## 4 Global atmospheric lifetime

While evaluation of the model results with satellite observations is useful, it only provides validation for altitudes seen by the instruments. The global atmospheric lifetime of a gas, on the other hand, probes the entire atmosphere and can thus be used as a diagnostic for the implementation of the $SF_6$ chemistry, as well as for the reanalysis data sets used in this work, by assessing the differences between the three simulations. The global atmospheric lifetime of species $i$ is calculated at any time from BASCOE model output as the ratio between the global atmospheric burden and the global atmospheric loss rate:

$$\tau = \frac{\text{atmospheric burden}}{\text{atmospheric loss rate}} \tag{2}$$

which can be found in, for example, the 2013 SPARC report on lifetimes, chapter 2 (SPARC, 2013). This is the instantaneous lifetime and depends on the specific atmospheric conditions. The atmospheric burden is calculated as the total number of molecules of the species $i$, in other words $\int n_i dV = \int \chi_i n_{air} dV$ where $n_i$ is the number density of species $i$, $n_{air}$ is the air

**Table 3.** BASCOE runs performed in this study

| Label | Description | Simulated period | Chemistry |
|---|---|---|---|
| TS_E5_L137 | Time slice simulation (perpetual QBO) driven by ERA5 on native grid | 20 years | no |
| TS_E5_L61 | Time slice simulation (perpetual QBO) driven by ERA5 on reduced grid with 61 levels | 20 years | no |
| TS_M2_L72 | Time slice simulation (perpetual QBO) driven by MERRA2 on native grid | 20 years | no |
| TS_M2_L49 | Time slice simulation (perpetual QBO) driven by MERRA2 on reduced grid with 49 levels | 20 years | no |
| TS_J3Q_L100 | Time slice simulation (perpetual QBO) driven by JRA-3Q on native grid with 100 levels | 20 years | no |
| TS_J3Q_L53 | Time slice simulation (perpetual QBO) driven by JRA-3Q on reduced grid with 53 levels | 20 years | no |
| TRANS_E5_L61 | Transient simulation driven by ERA5 on 61 levels | 1997/01 - 2023/12 | yes |
| TRANS_M2_L49 | Transient simulation driven by MERRA2 on 49 levels | 1997/01 - 2023/12 | yes |
| TRANS_J3Q_L53 | Transient simulation driven by JRA-3Q on 53 levels | 1997/01 - 2023/12 | yes |

density, $\chi_i$ is the volume mixing ratio of species $i$, and the integral runs over all grid cells of the model with volume $dV$. The atmospheric loss rate is a similar integral, but it includes the net chemical loss frequencies $\alpha_i$, so that the integral is given by $\int \alpha_i n_i dV$. The loss rate of species $i$ is written as:

$$\frac{dn_i}{dt} = -\alpha_i n_i \tag{3}$$

with

$$\alpha_i = L_i - \frac{P_i}{n_i} \tag{4}$$

where $P_i$ is the local production of species $i$ in molec/cm$^3$/s, and $L_i$ is the local loss in 1/s. These quantities $P_i$ and $L_i$ are computed by the chemical kinetic preprocessor and stored in the output files of BASCOE. Using equations (3) and (4) the global lifetime can be computed from every snapshot of model output. To smooth out seasonal variability, the lifetime is averaged over the period 2002-2012. This is an instantaneous lifetime, which can differ from the steady-state lifetime that is computed under equilibrium conditions, where the sinks and sources of the gas are balanced such that the burden remains constant. This condition is only fulfilled in specific simulations with constant reactant species. Other methods exist to compute the atmospheric lifetime from observations. See, for example, Plumb and Ko (1992), who developed a method to relate lifetimes of two species to the slope of the linear relation between the mixing ratios of two species in gradient equilibrium. The advantage of this method is that this can be done with observations of the extratropical lower stratosphere, but the drawback is that the lifetime of one of the two species must be known to derive a relative lifetime for the other one. In contrast, the burden divided by the loss, computed from model output, yields an absolute lifetime, but a limitation is that the uncertainties on the loss processes in models can be difficult to assess.

In the following, the lifetime is computed from model output for the six long-lived trace gases: $SF_6$, $N_2O$, $CH_4$, CFC-11, CFC-12 and HCFC-22. For $CH_4$ and HCFC-22, reaction with OH in the troposphere is the most important destruction process. Since BASCOE does not compute chemistry in the lower troposphere, only the stratospheric lifetime (and not the global lifetime) of these species is computed, i.e., integrating the loss rates above 200 hPa in the tropics (between $\pm\ 30°$) and above 400 hPa at other latitudes. The oceanic absorption of $N_2O$ is not taken into account in this work. This oceanic sink of atmospheric $N_2O$ is often overlooked. It is described as modest, but significant (de la Paz et al., 2025).

## 5 Results and discussion

### 5.1 Comparison with independent observations

The results of the BASCOE simulations have been compared to MIPAS and ACE-FTS satellite observations. The normalized mean difference (NMD) between the BASCOE simulation and the satellite profiles, computed as observation minus model divided by the model, the associated standard deviation (STD) and their correlations (Correl) are shown in Fig. 5 in the tropics and in Fig. 6 in the southern high latitudes (60°S-90°S). These statistics are computed over the period Jan. 2005 - Apr. 2012 for MIPAS and Feb. 2004 - Apr. 2012 for ACE-FTS. The grey shading indicates differences between MIPAS and ACE-FTS as discussed in Table 1. For $CH_4$ we used a conservative value of $\pm20\%$. For $SF_6$, the agreement between the BASCOE model runs and MIPAS is mostly within the instrument differences for MERRA2 and JRA-3Q, with ERA5 at the edge of the grey envelope. Agreement with ACE-FTS is within the uncertainties between 100 hPa and 20 hPa. The differences with both instruments are limited to approximately $\pm\ 10\%$. Note that instrument differences between ACE-FTS and MIPAS can reach up to $\pm10\%$ to $\pm20\%$ in certain regions, but this was not included in the envelope.

For $N_2O$ slightly larger differences are found between 10 and 50 hPa, but agreement is in line with instrumental differences below 50 hPa. For $CH_4$, the comparisons are within the range of uncertainty between MIPAS and ACE-FTS below 10 hPa and increase at high altitude where the chemical losses are more pronounced. For CFC-11 in the lower stratosphere, the difference is also in agreement with the instrumental differences and for HCFC-22 the difference is in agreement with the instrumental differences of around 14% at 25 km ($\sim 25$ hPa) found in Kolonjari et al. (2024). For CFC-12 the differences with ACE-FTS are somewhat larger than expected, showing differences of more than 20% above 30 hPa, which is larger than the 10% instrumental differences between MIPAS and ACE-FTS SPARC (2017). An explicit comparison between MIPAS and ACE-FTS is difficult here due to their different sampling. For $SF_6$ the simulation driven by MERRA2 shows smaller differences with respect to MIPAS than the simulations driven by ERA5 or JRA-3Q. For all species, the standard deviation is relatively small (<10%) at altitudes where their photochemical loss is small and increases at higher altitude. The correlation is also better in the lower stratosphere (>0.8) and decreases at higher altitude.

Figures 7 and 8 show time series of the normalized mean difference (NMD) and the normalized standard deviation (NSD) at different levels for $SF_6$. The time series in Fig. 7 emphasizes the temporal stability of the results in the tropics and at mid to low altitude. The difference is relatively stable and limited mostly to 10% in the tropics. At the Poles, the differences show

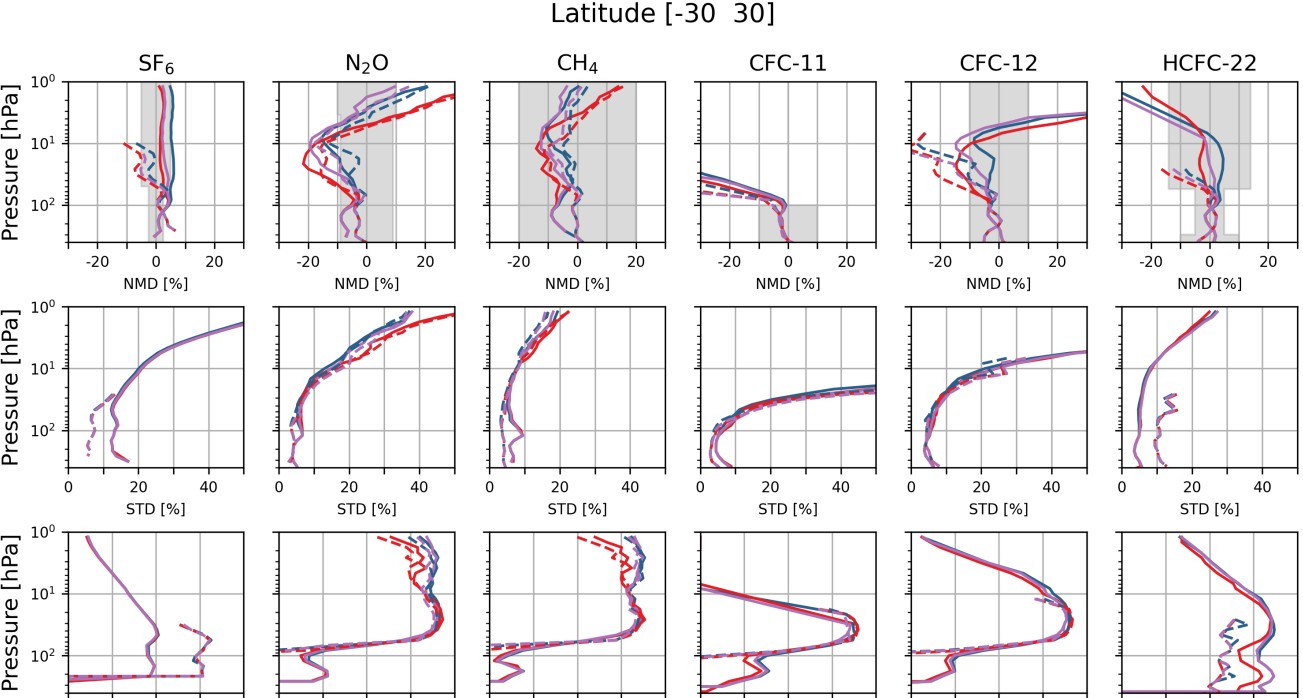

**Figure 5.** Normalized mean difference (observation-model)/model (NMD), standard deviation (STD) and correlation (Correl) of all six long-lived species over the tropics. Data is cut off at 1hPa because there are insufficient observations above this pressure level. The grey shading indicates typical differences between MIPAS and ACE-FTS.

seasonal variability with an amplitude around 20% at 1 hPa. The seasonality of the differences are also reflected in the standard deviation.

This oscillating pattern is found for all trace gases in this study (see the supplement where similar figures for the other long-lived tracers are available). The standard deviation is around 10-15% in the tropical lower stratosphere and increases towards the Poles.

The standard deviation shows a slight decreasing trend. This is due to the fact that the surface emissions of $SF_6$ significantly increase over the course of the shown period, thus reducing the relative noise in the observations. A similar pattern is found for HCFC-22 for the same reason (see Fig. S16).

## 5.2 Global atmospheric lifetime

Figure 9 shows the global atmospheric lifetimes from the three BASCOE runs and the multi-reanalysis mean (MRM), averaged over 2002-2012. The error bars indicate the $1\sigma$ variability of the lifetime time series from BASCOE over the considered period

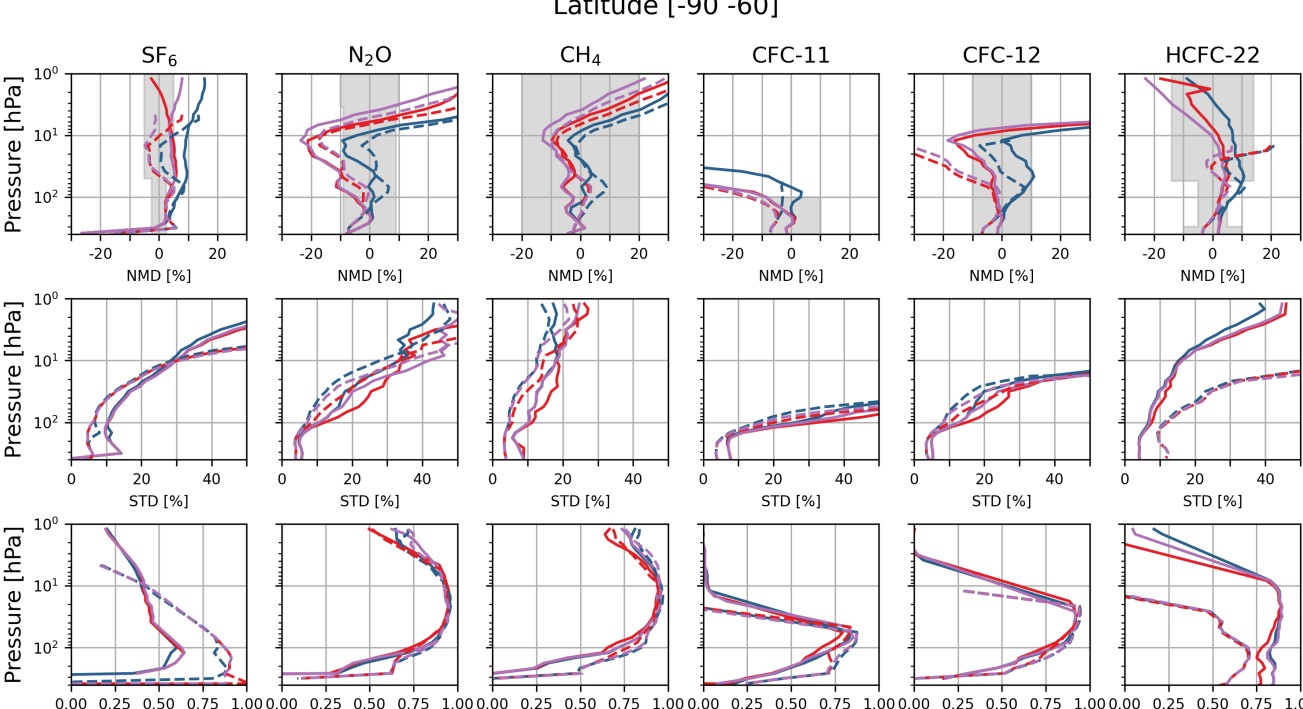

**Figure 6.** Same as Fig. 5, but for the South Pole.

2002-2012. The results are compared with lifetimes found in the literature, shown as error bars here to indicate the possible range of values with a marker at the best value if provided in the source paper. Note that these reference values are not specifically for the period 2002-2012 and that the error bars represent different quantities and are not actual uncertainties. Note

also that some lifetime estimates are model based, while other lifetimes are derived from observations. Model based lifetime estimates usually rely on the assumption of a steady-state atmosphere. Minschwaner et al. (2013) computed a correction factor for transient lifetimes of CFC-11 and CFC-12 and showed that the difference between transient and steady-state lifetimes was on the order of 1-2% for these species. This correction factor takes into account the time it takes for tropospheric increases or decreases to propagate to the loss region in the stratosphere. The lifetimes of the long-lived species $N_2O$, $CH_4$, CFC-

11, CFC-12 and HCFC-22 (see Fig. 9a) show good agreement with results from the literature (SPARC, 2013; Volk et al., 1997; Prather et al., 2023; Fleming et al., 2015; Minschwaner et al., 2013; Moore and Remedios, 2008; Avallone and Prather, 1997; Kanakidou et al., 1995; Spivakovsky et al., 2000) and between the reanalyses. For $CH_4$ and HCFC-22, only stratospheric lifetimes are shown. Note that none of the $N_2O$ lifetimes presented here take into account the uptake by oceans. We hypothesize that the true lifetime of $N_2O$ is thus smaller than what is found here. Our lifetime of $CH_4$ is remarkably higher than that of

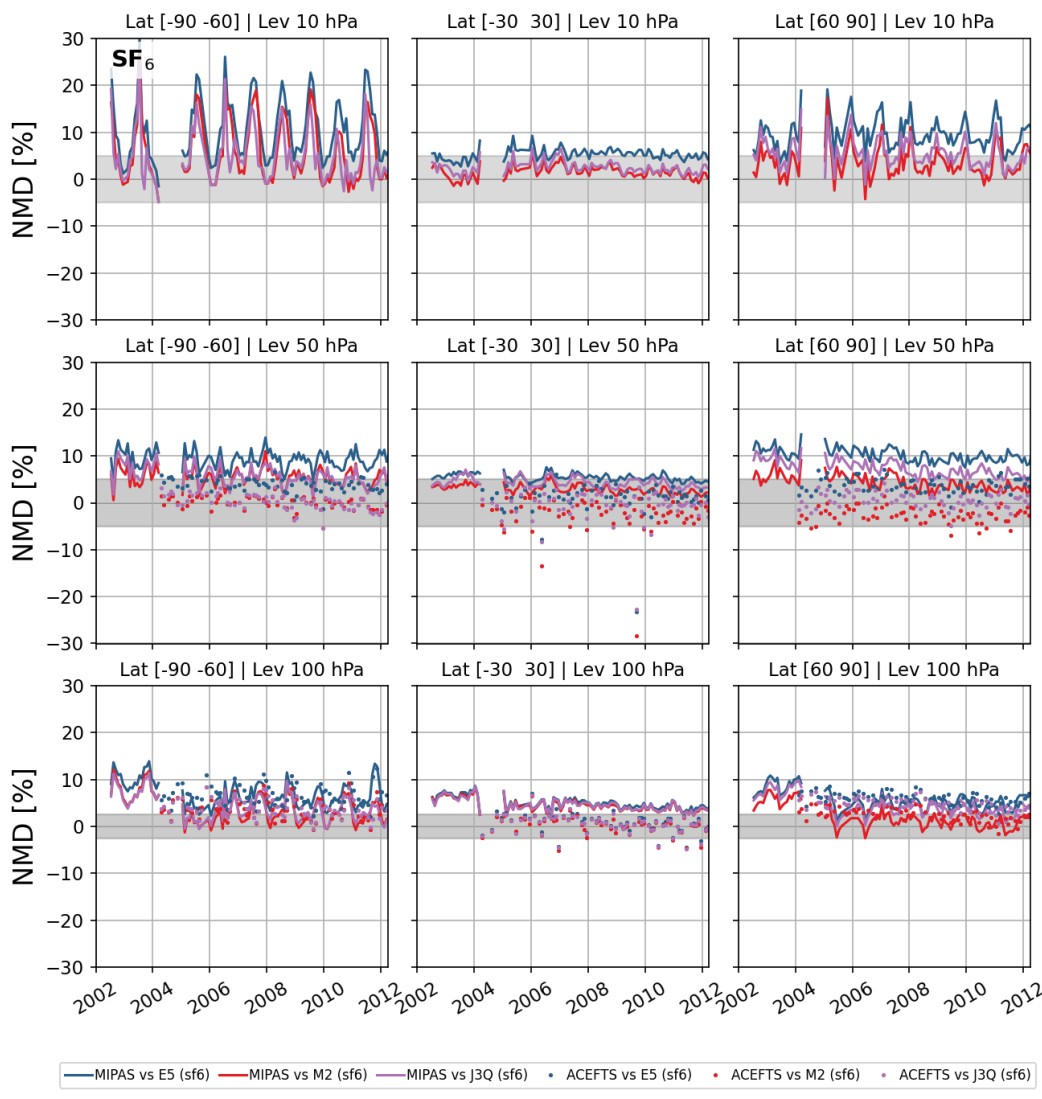

**Figure 7.** Timeseries of the normalized mean difference of $SF_6$ volume mixing ratios with respect to MIPAS and ACE-FTS for three latitude bands and three pressure levels. Grey shading indicates the uncertainties between MIPAS and ACE-FTS from validation studies.

Volk et al. (1997), which could be due to uncertainties in the OH concentrations. We also find a lower HCFC-22 lifetime than Moore and Remedios (2008), which may similarly reflect differences in modelled OH concentrations.

The consistency between the three different simulations for the 5 species mentioned above and the agreement with the literature creates confidence in our lifetime computation. While these species have much shorter lifetimes than $SF_6$ and have

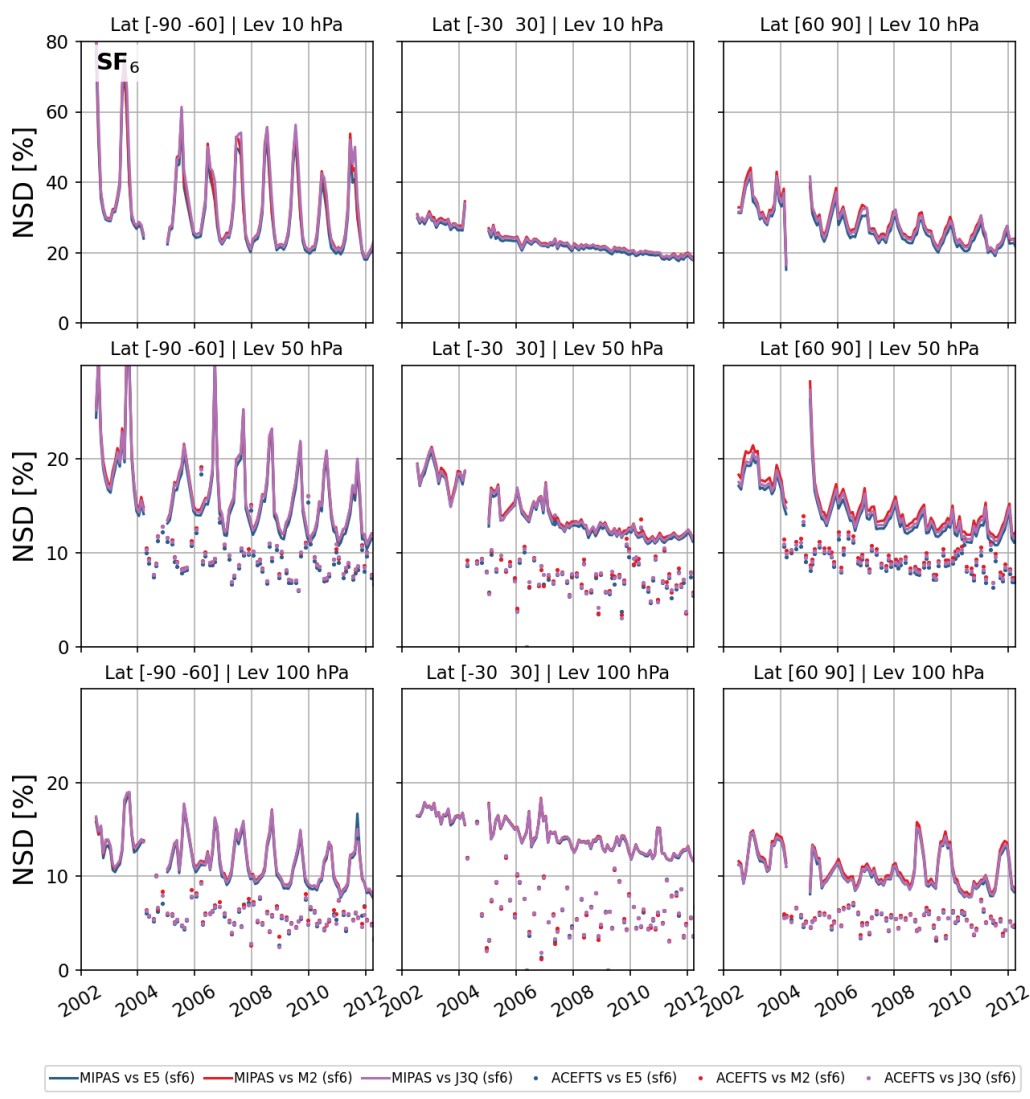

**Figure 8.** Timeseries of the standard deviation of $SF_6$ for three latitude bands and three pressure levels.

sinks in different regions, the diagnostic used here is independent of the absolute lifetime for species with well-defined loss

processes. A limitation of the validation of the lifetime computation is that losses outside of the model domain are not taken into account. However, BASCOE reaches altitudes of around 80 km where the maximum loss of $SF_6$ occurs (Kouznetsov et al., 2020). The lifetimes of all six species are summarized in Table 4. The global atmospheric lifetime of $SF_6$ has been computed in several studies over the last 30 years (Krey et al., 1977; Ravishankara et al., 1993; Ko et al., 1993; Morris et al.,

1995; Harnisch et al., 1999; Reddmann et al., 2001; Patra et al., 1997; Ray et al., 2017; Kovács et al., 2017; Kouznetsov
et al., 2020; Loeffel et al., 2022). This body of work is summarized in Fig. 9b. The black bars are organized (from left to
right) from the oldest to the most recent publication. Some of these estimates were derived from models, while others were
derived from observations so that this panel does not represent an evolution in model development. Despite the number of
estimates, the atmospheric lifetimes show a large spread. Ravishankara et al. (1993) obtained a lifetime of 3200 years from
model computations considering only UV-photolysis and a slow electron attachment that leads to destruction of $SF_6$. The lower
limit mentioned there was obtained by maximizing removal by combustion. Later modelling studies like Morris et al. (1995)
reduce the lifetime of SF6 from 3200 years to around 800 years by considering also a fast, non-destructive electron attachment
branch. In the present study, the average lifetime of $SF_6$ in BASCOE over the period 2002-2012 is $2646 \pm 532$ $(1\sigma)$ years with
an ERA5 driven run, $1909 \pm 182$ $(1\sigma)$ years with a MERRA2 driven run and $2145 \pm 459$ $(1\sigma)$ years with a JRA-3Q driven run.
The uncertainty on the lifetime coming from the choice of reanalysis can also be estimated as the root of the squared differences
with respect to the MRM. This yields an uncertainty of 532 years, similar to the standard deviation over 10 years. Given the
large differences in seasonal variability for $SF_6$ lifetime between the three reanalyses (see Fig. 10) the results have been split
up into the averages of the maxima (equinoxes) and minima (solstices) of the time series. The BASCOE averages show good
agreement with the result from the transient reference simulation (REF) and the simulation nudged to ERA-Interim in Loeffel
et al. (2022), using the global chemistry climate model EMAC. This is especially true for the simulation driven by JRA-3Q.
The lifetime from the simulation driven by MERRA2 is similar to the lifetime from the time slice simulation (TS2000) in
Loeffel et al. (2022), which uses fixed climate conditions from the year 2000. The lifetime derived from the ERA5-driven
simulation is closer to their CSS simulation, which uses the same climate conditions as the transient reference simulation, but
sets the concentrations of the $SF_6$ reactant species to 1950 levels. Due to the considerable tropospheric growth rates of $SF_6$ of
around 5% year$^{-1}$, the instantaneous lifetime of this species is expected to be significantly larger than the steady-state lifetime.
We compute a correction for the instantaneous lifetime of $SF_6$, using the formula by Minschwaner et al. (2013) mentioned
above, assuming an age of air of 8 years in the region of maximum loss. This gives an estimate for the steady-state lifetime that
is 20-30% shorter than the instantaneous lifetime. Additionally, a one day simulation was done to assess the sensitivity to the
volume mixing ratios of $SF_6$ on the lifetime. Given that BASCOE slightly underestimates the $SF_6$ concentrations (see Fig. 5
and 6), we increased the initial conditions of $SF_6$ by 10% and ran the model for one day to analyse the impact of the volume
mixing ratios. We found that the instantaneous lifetime of one snapshot of the model decreased by about 30 years. Similarly, if
the volume mixing ratios were decreased by 10%, the lifetime increased by about 40 years. This difference is small compared
to the total lifetime, on the order of 1%, which indicates that the chemistry of $SF_6$ is more or less linear, i.e., not highly coupled
with other gases. This implies that the lifetime computation is relatively independent of potential errors in the volume mixing
ratios.

For $N_2O$, $CH_4$, CFC-11, CFC-12 and HCFC-22, the lifetimes derived from the three different simulations are closely aligned
(see Fig. 9a), indicating that there is little sensitivity to meteorology for these species, possibly due to good agreement between
the reanalyses at altitudes where the species are destroyed. However, TRANS_E5_L61 consistently produces a slightly lower
lifetime than TRANS_M2_L49, TRANS_J3Q_L53 being often in between. For $SF_6$ there are relatively large differences

between the three values. This could be explained by the lack of stratospheric and mesospheric observations to constrain the reanalyses in the region where $SF_6$ chemistry is important, as well as the differences in the sponge layers between the three reanalyses. Also, contrary to the 5 other species, $SF_6$ lifetime from TRANS_E5_L61 is significantly longer than the lifetime from TRANS_J3Q_L53 and TRANS_M2_L49. We hypothesize that $SF_6$ spends less time in the mesosphere due to the faster circulation with ERA5 and is therefore able to return to the stratosphere via recirculation, resulting in higher volume mixing ratios in the stratosphere and thus a longer lifetime.

The time series of the lifetime of all six species and the tropical temperature at 1 hPa are shown in Fig. 10. There is an increasing trend in the lifetime of $CH_4$, consistent with predictions from Li et al. (2022), but not for the other species, while Prather et al. (2023) found a decreasing trend of $N_2O$ lifetimes. There is no clear trend in the $SF_6$ lifetime despite the strong growth of $SF_6$, in contrast to the increasing trends in the reference simulation and the projected simulation from Loeffel et al. (2022). Instead, the lifetimes are steady over the entire simulation range, similar to the time-slice simulation in Loeffel et al. (2022), which was used to compute a steady-state lifetime. In the time series, a semi-annual oscillation (SAO, Reed, 1965) is observed. The SAO is more apparent and more regular for $CH_4$ and HCFC-22 than for CFC-11, CFC-12 and $N_2O$. SAO's in the winds and temperature have been previously observed near the stratopause around 1 hPa and near the mesopause at around 0.01 hPa (Shangguan and Wang, 2023). The SAO in the stratosphere is thought to be driven by Kelvin waves, caused by convection in the tropics, and linked to the passage of the Sun across the equator that happens twice per year and the absorption of solar UV by ozone in the stratosphere. The origin of the mesospheric SAO is not well understood, but it is considered that the mesospheric SAO is driven by selective transmission of inertia-gravity waves and there appears to be a lag between the stratospheric and the mesospheric SAO (Richter and Garcia, 2006). The temperature is plotted in the tropics at 1 hPa, the region where the SAO is the strongest (Shangguan and Wang, 2023). While similar patterns are observed between the temperature at 1 hPa in the tropics and the time series of the lifetime, the link between both quantities is not clear. For $SF_6$, the amplitude of the seasonal variation of the lifetime is larger for TRANS_E5_L61 than for the two other simulations. This could be related to the stronger seasonal cycle in the total tropical upwelling of ERA5 with respect to MERRA2 (SPARC, 2022, Chap. 5). For $SF_6$ the maxima in the lifetime happen in spring and autumn, around the equinoxes, while the minima are in summer and winter around the solstices. For $N_2O$ the occurrence of peaks and valleys is reversed, with the maxima around the solstices and the minima around the equinoxes. $CH_4$ and HCFC-22 are similar to $SF_6$, while CFC-11 and CFC-12 are more similar to $N_2O$.

**Table 4.** Average instantaneous lifetime (years) of six long-lived trace gases from the different BASCOE simulations for the period 2002-2012 and their $1\sigma$ standard deviation. For $CH_4$ and HCFC-22, stratospheric lifetime is shown, while global atmospheric lifetime is shown for the other species.

| Experiment | $N_2O$ | $CH_4$ | CFC-11 | CFC-12 | HCFC-22 | $SF_6$ | $SF_6$ max | $SF_6$ min |
|---|---|---|---|---|---|---|---|---|
| TRANS_E5_L61 | $130 \pm 10$ | $144 \pm 7$ | $52 \pm 5$ | $108 \pm 9$ | $198 \pm 11$ | $2646 \pm 532$ | $3412 \pm 286$ | $2058 \pm 133$ |
| TRANS_M2_L49 | $135 \pm 14$ | $147 \pm 8$ | $59 \pm 7$ | $115 \pm 14$ | $204 \pm 12$ | $1909 \pm 182$ | $2079 \pm 149$ | $1755 \pm 136$ |
| TRANS_J3Q_L53 | $135 \pm 11$ | $142 \pm 7$ | $56 \pm 5$ | $114 \pm 10$ | $195 \pm 11$ | $2147 \pm 459$ | $2741 \pm 237$ | $1660 \pm 217$ |
| TRANS_MRM | $133 \pm 11$ | $144 \pm 7$ | $56 \pm 5$ | $112 \pm 11$ | $199 \pm 11$ | $2235 \pm 372$ | $2717 \pm 183$ | $1802 \pm 123$ |

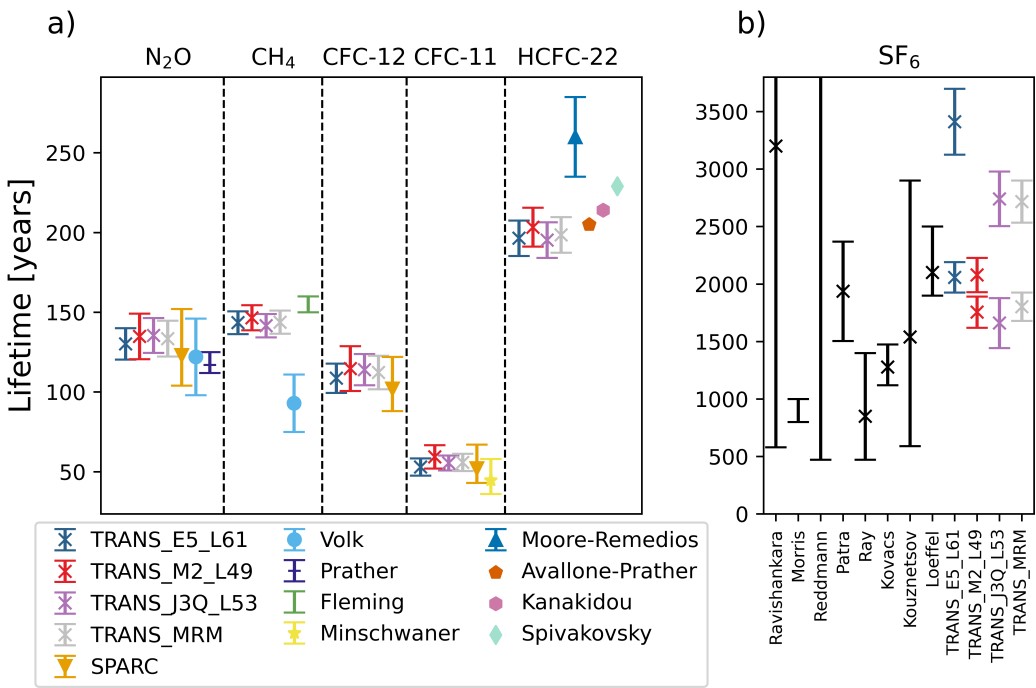

**Figure 9.** Lifetime of atmospheric trace gases averaged over 2002-2012. The $SF_6$ lifetimes from the literature are shown as error bars without marker point if only a range was given, or as an error bar with a marker at the most likely lifetime value, or as a single point if only one value was given. The lifetimes for $CH_4$ and HCFC-22 are integrated only over stratospheric levels (see methods)

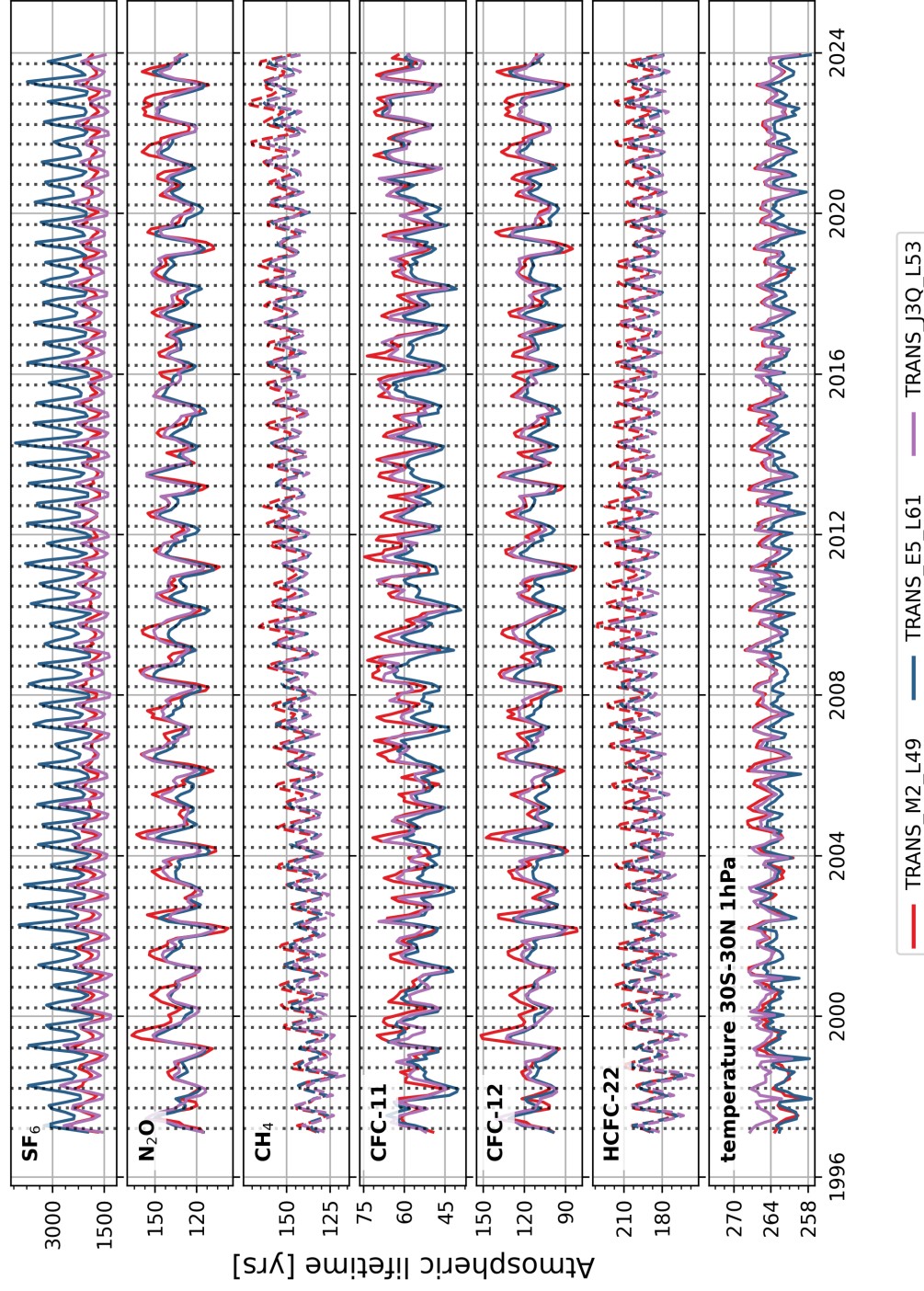

**Figure 10.** Global atmospheric lifetime (solid lines) or stratospheric lifetime (dashed lines) of different species computed from BASCOE simulation drived by ERA5 (blue lines), MERRA2 (red lines) and JRA-3Q (purple lines).

## 6 Conclusions

The chemistry of $SF_6$ has been implemented in the Belgian Assimilation System for Chemical ObsErvations (BASCOE). Three model simulations have been carried out, driven by three recent meteorological reanalyses (ERA5, MERRA2 and JRA-3Q) that include the mesosphere where $SF_6$ is destroyed via auto-attachment with electrons. These simulations show a relatively good agreement with MIPAS and ACE-FTS satellite observations in the middle and low stratosphere with differences generally within $\pm 10\%$ for $SF_6$ volume mixing ratios and mostly within the estimated instrumental differences for all six species below 10 hPa. For $SF_6$, the simulation driven by MERRA2 has a slightly smaller differences with respect to MIPAS than those driven by ERA5 and JRA-3Q. The instantaneous global atmospheric lifetime of $SF_6$ was computed and compared with published results to further assess the impact of the choice of meteorology in the mesosphere where satellite observations provide little information. The lifetime was also computed for the other long-lived species. The computed lifetime of $SF_6$ - 2646 $\pm$ 532 (1 $\sigma$) years (ERA5), 1909 $\pm$ 182 (1 $\sigma$) years (MERRA2) and 2147 $\pm$ 459 (1 $\sigma$) years (JRA-3Q) - is consistent with recent results by Loeffel et al. (2022), but demonstrates the sensitivity of $SF_6$-derived transport diagnostics to the underlying meteorological data. For the other trace gases, the lifetimes from the three simulations are in good agreement between themselves and with published results, which confirms the validity of our lifetime calculation. A semi-annual oscillation is observed in the time evolution of the lifetimes of all six species, which exhibits similar patterns as tropical stratospheric temperature variations. A follow-up study is necessary to further investigate the differences in stratospheric transport among the three reanalyses in order to explain the differences in seasonal variation observed in the lifetime. With this model update, BASCOE is ready to perform further transport studies based on $SF_6$ diagnostics.

## Appendix A: Computation of the reduced grids

This appendix describes the procedure to construct a vertical grid with reduced resolution for the reanalyses. The reanalyses used in this work are defined on a hybrid sigma-pressure grid. The pressure at level interface $p_{mid}(i)$ is defined by the model level parameters $A_p(i)$ and $B_p(i)$ as follows:

$$p_{mid}(i) = Ap(i) + Bp(i) \cdot p_{surf} \tag{A1}$$

with $p_{surf}$ being the surface pressure and $i = 1, \mathrm{nlev}$ is the level index and nlev is the number of model levels. The reduced grids are obtained as follows, see also the illustration in Fig. A1:

– Step 1: For several altitudes between 5 and 90 km, values of the desired vertical resolution dz were provided, hereby aiming to stay close to the native resolution in the upper levels and reducing the resolution in the troposphere.

– Step 2: These proposed altitude-resolution points were then fitted with a polynomial.

– Step 3: Starting from the surface, the vertical resolution at the lowest level is calculated using this polynomial, and this defines the altitude of the lowest level. Then level i+1 was calculated by incrementing level i with the vertical interval dz from the polynomial at level i, ensuring that the top levels correspond to the top levels of the reanalysis.

– Step 4: Then the native $A_p$ and $B_p$ are interpolated to the new vertical grid using a log-pressure approximation for the altitude. From the new $A_p$ and $B_p$ coefficients, the sigma-pressure levels were computed.

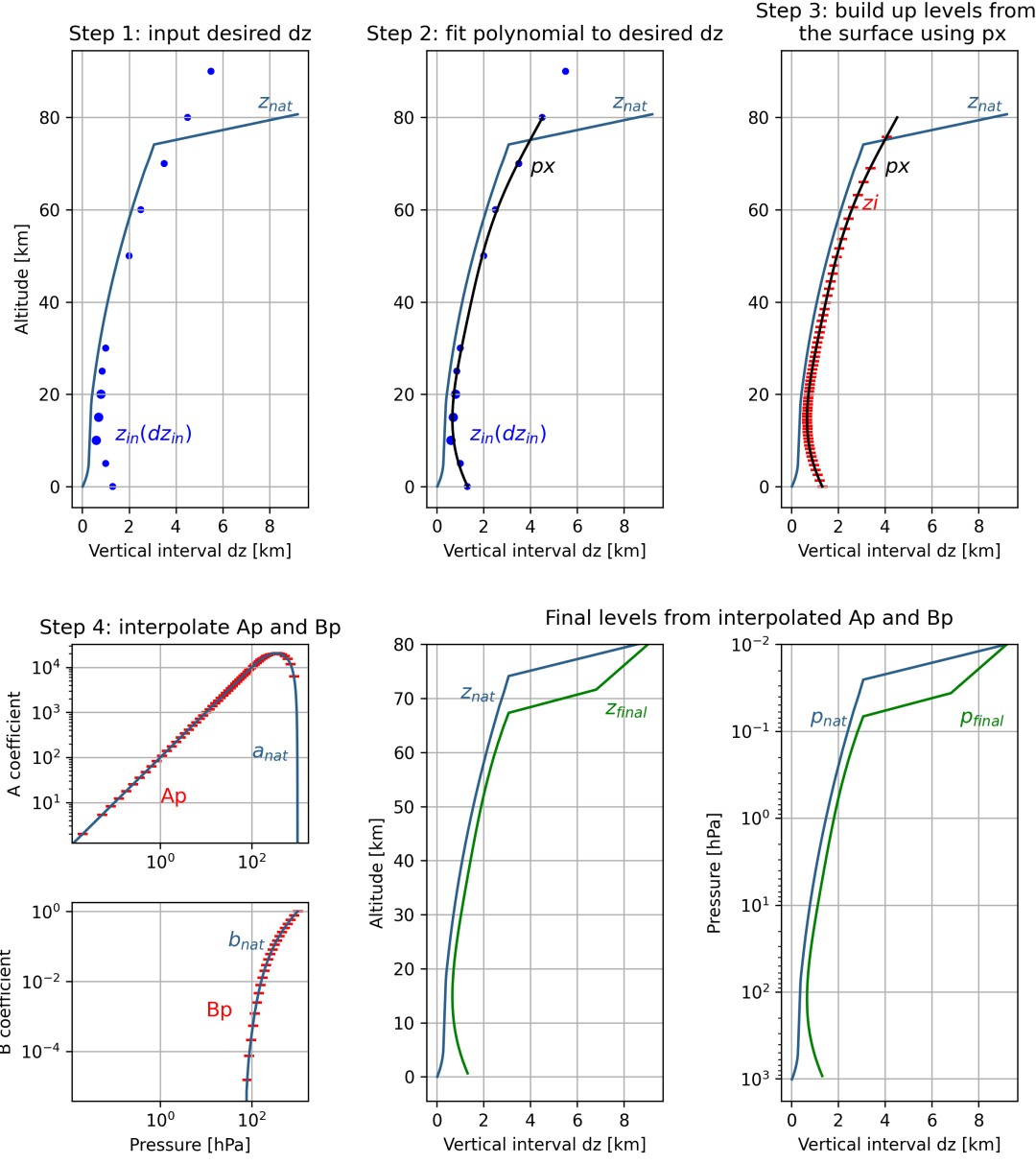

**Figure A1.** Workflow to construct a new grid for the reanalyses with desired vertical resolution (see text for details).

*Code availability.* The BASCOE code is available upon request.

*Data availability.* The MIPAS data is available at https://www.imk-asf.kit.edu/english/308.php. The ACE-FTS data is available at https:
390 //uwaterloo.ca/atmospheric-chemistry-experiment/. Monthly zonal mean BASCOE output is available at https://doi.org/10.5281/zenodo.
17751040.

*Author contributions.* SV executed the implementation of the chemistry, carried out the simulations and visualized and interpreted the results. QE and EM provided the idea for the study, supervised the work and contributed to the interpretation of the results and revising of the manuscript. EM is senior research associate with F.R.S.-FNRS". SC and MOB helped with the interpretation of the results and provided
feedback on the manuscript. TR provided support in the implementation of $SF_6$ chemistry in BASCOE and provided feedback on preliminary results. GS provided support in the use of MIPAS data sets. RE provided the EMAC data set and provided feedback on preliminary results.

*Competing interests.* At least one of the authors is a member of the editorial board of Atmospheric Chemistry and Physics.

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
