# Peer review of "Atmospheric lifetime of sulphur hexafluoride ( $\text{SF}_6$ ) and five other trace gases in the BASCOE model driven by three reanalyses"

_EGUsphere, 2025_

## Author Comment (AC1)

*We thank the reviewer for their encouraging assessment of the manuscript. The reviewer's comments are given below in* black *with our responses in blue italics.*

This paper describes the results of an update to the BASCOE model to include detailed SF6 chemistry in the mesosphere. The lifetime of SF6 as well as of five other trace gases are calculated from the model output and they all generally agree well with previous lifetime estimates. This suggests that the SF6 chemistry implemented in BASCOE is at least sufficient to represent the main loss processes of SF6 and that it is then primarily the model transport that will determine the SF6 lifetime. The SF6 lifetime is shown to vary considerably with different reanalysis data sets driving the model but the lifetimes still agree within uncertainties. The accurate representation of SF6 chemistry in BASCOE is an important advancement in assessing model transport compared to observations.

The paper is well written and the results are described clearly. The topic is appropriate for ACP and so I suggest publication with consideration of the minor comments listed below.

Minor comments:

Line 24: I would suggest replacing 'General' with 'Chemistry-'.

*We replaced "General climate models" by "Climate models" as not only chemistry-climate models predict this increase.*

Table 1: The entries in this table are somewhat confusing and repetitive. It doesn't seem like you need the fifth column with the Data Versions since they're all the same. Maybe just state the versions in the table header. Also, the 'Agreement' and 'Notes and References' columns sometimes overlap in their content and frequently include the subjective term 'Good' that isn't necessarily helpful. The 'Compared instruments' and Data versions' columns seem to have conflicting versions. For instance, for SF6 the ACE V2.2 and V5.3 are listed. I would suggest trimming this table down to the basic information and make sure it isn't repetitive or conflicting.

*We use the validation studies that are available as a reference for our own results. To our knowledge, there are no validation studies for SF6 with a more recent version of ACE-FTS. The subjective statements have been removed, and the table has been simplified.*

Line 131 'parameterized' is misspelled

*We use Oxford spelling throughout the article. Parametrized is an accepted British English spelling. We have changed "sulfur hexafluoride" to "sulphur hexafluoride" to be more consistent with Oxford spelling.*

Fig. 7: The y-axis scales here seem much too large, it's difficult to see any features. Maybe that's the point but it seems like you could at least go to +/-30%. It also might be helpful to indicate the instrument differences for each species by dashed lines for instance.

*Done.*

---

## Author Comment (AC2)

*We thank the reviewer for their time and efforts to write a thorough review of the manuscript. The reviewer's comments are given below in* black *with our responses in blue italics.*

This manuscript presents the implementation of SF6, an important species to indirectly help constrain stratospheric circulation speeds, into the BASCOE model. An important result is the dependency of the model-based lifetime estimate of any trace gas on the meteorological reanalysis used. A lot of work, both technical and scientific, appears to have gone into the development of the model and its setups. This is very commendable. At the same time, the manuscript is somewhat light on the discussion of the results and their placing in a wider context, particular when it comes to the interpretation of differences with observations and other studies. Section 5 lacks both depth and detail in its current state, particularly also with regard to the actual uncertainties of the lifetimes derived. Once these issues, as well as the more detailed ones listed below, have been addressed, the manuscript could be published in ACP.

*In addition to the updates in response to your specific comments, the revised manuscript provides a deeper discussion in Sect. 5 where we tried to make the distinction clear between the different kinds of lifetime estimates (see replies to specific comments below). Some differences were pointed out between our lifetime estimates and the literature for a more balanced discussion of Fig. 9. We have also tried to assess the uncertainties on the lifetimes, even though uncertainties on the loss rates in the model are hard to derive based on differences between the model simulations and observations. Furthermore, the revised manuscript provides the lifetime of the multi-reanalysis mean, and the large seasonality in the lifetime time series of SF6 is emphasized by looking at the lifetime average of the maxima and minima separately.*

l16-17 Consider revising this statement to make it less absolute. Not all studies of middle atmospheric transport have been "motivated by the existence of the stratospheric ozone layer".

*"Studies of middle atmospheric transport have been important for a long time and are motivated by the existence of the stratospheric ozone layer" changed to "Studies of middle atmospheric transport have been important for a long time and are motivated **in part** by the existence of the stratospheric ozone layer"*

l17-19 Transport in the mesosphere (which is part of the middle atmosphere) is not dominated by the stratospheric BDC, nor is isentropic transport in the LMS. Consider revising this statement accordingly.

*"Transport of trace gases in the middle-atmosphere is dominated by a large scale circulation pattern in the stratosphere, called the Brewer-Dobson circulation (BDC)" changed to "Transport of trace gases in the stratosphere is dominated by a large scale circulation pattern that is part of the Brewer-Dobson circulation (BDC)"*

l24-27 It seems a bit of a stretch to use publications that are 16-24 years old for a claim on what climate models predict.

*These are commonly cited papers on this topic which are still valid. We have kept one of these older works and added two more recent ones. "Butchart and Scaife, 2001; Butchart et*

*al., 2006; Garcia and Randel, 2008, Calvo and Garcia, 2009; McLandress and Shepherd, 2009" changed to "Butchart et al., 2006; Hardiman et al., 2013; Abalos et al., 2021"*

l28-29 Recommend modifying this to "Changes in AoA at a given location in the stratosphere thus reflect changes in circulation speed" or similar.

*Suggestion accepted.*

l30-31 Two-way mixing is not correctly represented here (see, e.g., Garny et al., 2014)

*Changed "Additionally, two-way mixing can increase stratospheric AoA, partly through making air parcels recirculate" to "Additionally, two-way mixing between the tropics and the extra-tropics can increase stratospheric AoA by making air parcels recirculate, while in the extra-tropical lower stratosphere, mixing rejuvenates the air. Mixing within the extra-tropical stratosphere also has local effects and flattens the latitudinal age gradient".*

l31 Three incomplete references.

*Fixed.*

l31-32 This is a bit confusing. Why are four references required for a definition of AoA?

*We removed two references. We kept one old and one new.*

l34-35 SF6 has its weaknesses, too. A more balanced discussion is recommendable here.

*Indeed, the limitations should be mentioned here. We have added a brief discussion of the mesospheric sink and the noisiness of the satellite data products, as well as solutions that address these issues.*

l38 Climate studies?

*Changed "climatological studies" to "climate studies".*

l39-40 It would be useful to have a bit more detail on the nature and scale of these discrepancies.

*We have added a few sentences about the magnitude of the trends and the differences in age of air and cite Garny et al., Rev. Geophys., 2024.*

l42-44 Please add references for the two satellite instruments and the three reanalyses.

*Done.*

l70-85 The definition of IMK/IAA is missing. Also, it is not clear (here, in the table, and throughout the manuscript) what is meant by bias. Is this the actual accuracy of the instruments, and if yes, against which "ground truth"? Or would perhaps "difference" be the more appropriate term?

*We have added the definition of IMK/IAA in the subsection that describes the MIPAS data. Indeed, bias can be a misleading term here. We have replaced the word "bias" by*

*"difference", as it refers to differences between instruments or between CTM output and satellite observations, where there is no ground truth against which the volume mixing ratios are evaluated.*

l82 Looking this up in Saunders et al. 2025, ACE SF6 in V5.2 is problematic. Has this issue been resolved in V5.3?

*In V5.3 the number of spikes around 30km in the VMR profiles was reduced.*

l105 According to Ploeger et al., ACP, 2021, ERA-5 also has a representation of the BDC that is likely too slow.

*There is no direct comparison of the age of air from ERA5 and MERRA2. Ploeger et al. 2021 indeed found a slow circulation compared to ERA-Interim and similarities in tropical upwelling and the evolution of residual circulation transit times. We have added a sentence to nuance this.*

l115 This sentence appears to suggest that Burkholder et al., 2015 is the most recent version, which is not the case.

*Some reaction rates can be updated with respect to Burkholder et al. (2015). None of the updated reactions are important for the chemistry of the six trace-gases studied in this work.*

l128 Why were the rate constants taken from a modelling paper and not from kinetic studies (such as Burkholder et al. 2015)?

*The JPL evaluations do not include ion reactions, which are crucial for SF6 chemistry.*

l134 Please explain why electrons from the ionosphere are relevant for mesospheric chemistry.

*The D-region of the ionosphere is located in the mesosphere, between 60 km and 90-95 km. We have added this information.*

Table 2 Consider adding "in the mesosphere" to the caption.

*As some of the reaction rates are non-negligible at altitudes below 50 km, around the lower boundary of the mesosphere, we choose not to add "mesosphere" to the caption. See Fig. 5 in Reddmann et al., J. Geophys. Res., 2001.*

Figure 1 Longitude should be E or W, not N? Also, according to the literature (e.g., Ravishankara et al., 1993 or Morris et al., 1995), the electron energy plays an important role for the relative importance of the individual SF6 reactions. How was this taken into account here?

*The longitude was corrected.*

*We use the electron attachment rate from Reddmann et al., J. Geophys. Res., 2001 for associative attachment, which is specific for thermalized electrons (<<1eV). Thermalization is a very fast process at these altitudes, justifying the use of the electron density computed*

*from balance between ionization and recombination.*
*There is also a dissociative attachment reaction with a branching ratio of less than 0.001,*
*which we neglect. This reaction only becomes relevant above 100km, as seen in Fig. 9 in*
*Totterdill et al., J. Phys. Chem. A (2015).*
*We have added the following sentence: "We use an electron-attachment rate that is specific*
*for thermalized electrons (<< 1 eV), resulting in fast attachment. As thermalization is fast at*
*the altitudes relevant for electron-attachment, the use of the full electron density is justified."*
*as well as the altitude range of the D-region.*

l157-158 This statement does not seem to be correct for the ERA-5 vertical resolution.

*Changed "From this figure it is clear that the resolution is now lower in the troposphere and,*
*to a lesser extent, in the stratosphere and mesosphere." to "From this figure it is clear that*
*the resolution is now lower in the troposphere, thus removing unnecessary levels, while also*
*being somewhat reduced in the stratosphere and mesosphere."*

l175 "Spatial grid" instead of "space"?

*Suggestion accepted.*

l199-202 A similar constraint applies to the oceanic lifetime of N2O. Considering this may
affect some of the later results and discussion.

*Thank you for bringing this to our attention. At present we do not consider the tropospheric*
*sink of N2O. The values to which our estimates are compared also do not account for this*
*oceanic sink.*

l238-240 This is a repetition from section 4.

*Changed "Since CH4 and HCFC-22 have significant sinks in the troposphere, and since*
*BASCOE is not designed for tropospheric chemistry, the stratospheric lifetime of these*
*species has been computed and compared with stratospheric lifetimes from the literature." to*
*"For CH4 and HCFC-22, only stratospheric lifetimes are shown."*

l245 Figure 9 represents an apples-and-oranges comparison as some of the lifetimes in
there are equilibrium lifetimes and some are not. Given the substantial growth rates of the
mole fractions of some of the species considered here (especially SF6), the equilibrium
lifetime will be significantly different from the instantaneous one (see, e.g., SPARC, 2013).
Not considering this is a major shortcoming of this work.

*It was shown that for some species, like CFC-11 and CFC-12, the transient lifetime is very*
*similar to or evolves towards the steady-state lifetime (SPARC 2013). For SF6 we have*
*estimated a correction factor to convert the instantaneous lifetime to a steady-state lifetime*
*using a formula developed in Minschwaner et al., ACP (2013). We have added several*
*sentences to clarify the differences between the lifetimes shown and have used this*
*correction formula by Minschwaner et al., ACP (2013) to compute a correction on the*
*instantaneous lifetime of SF6.*

l246-248 This is a very short summary of the state of the art of SF6 lifetime estimates. It
does not take into account (or discuss) the wide range of approaches (satellite and in situ

observations as well as model-based approaches) and their strengths and weaknesses, nor does it consider the advances that have been made since 1993.

*Changed "The values show a large spread. Ravishankara et al. (1993) proposed a value of 3200 years from model computations, but noted that taking into account electron attachment could significantly reduce the lifetime, hence the lower limit of 580 years. In later works, the importance of the mesospheric sink is generally recognised." to "Despite the number of estimates, the atmospheric lifetimes show a large spread. Ravishankara et al. 1993 obtained a lifetime of 3200 years considering only UV-photolysis and a slow electron attachment that leads to destruction of $SF_6$. The lower limit mentioned there was obtained by maximizing removal by combustion. Later modelling studies like Morris et al. 1995 reduce the lifetime of $SF_6$ from 3200 years to around 800 years by considering also a fast, non-destructive electron attachment branch."*
*A few sentences have also been added in section 4 to differentiate between the different methods to compute lifetimes and their strengths and weaknesses: "This is an instantaneous lifetime, which can differ from the steady-state lifetime that is computed under equilibrium conditions, where the sinks and sources of the gas are balanced such that the burden remains constant. Other methods exist to compute the atmospheric lifetime from observations. See, for example, Plumb and Ko (1992), who developed a method to relate lifetimes of two species to the slope of the linear relation between the mixing ratios of two species in gradient equilibrium. The advantage of this method is that this can be done with observations of the extratropical lower stratosphere, but the drawback is that the lifetime of one of the two species must be known to derive a relative lifetime for the other one. In contrast, the burden divided by the loss, computed from model output, yields an absolute lifetime, but a limitation is that the uncertainties on the loss processes in models can be difficult to assess."*

Figure 7 Why is the scale on the y-axis so large? It results in most of the figure showing empty space.

*Reduced to -30% to 30% and added grey shading for the instrument uncertainties.*

l268 The more recent Prather et al., ACP, 2023 might be a better reference here.

*Indeed, this was a typo.*

l275 Consider rewording to clarify whether the mesospheric SAO is dependent on gravity or gravity waves.

*The driving of the SAO has been clarified and a reference was added. We changed "The origin of the mesospheric SAO is not well understood, but there is thought to be a lag between the stratospheric and the mesospheric SAO, and that the mesospheric SAO is related to gravity and Kelvin waves." to "The SAO in the stratosphere is thought to be driven by Kelvin waves, caused by convection in the tropics, and linked to the passage of the Sun across the equator that happens twice per year and the absorption of solar UV by ozone in the stratosphere. The origin of the mesospheric SAO is not well understood, but it is considered that the mesospheric SAO is driven by selective transmission of inertia-gravity waves and there appears to be a lag between the stratospheric and the mesospheric SAO (Richter and Garcia, 2006)."*

Table 4 The uncertainties listed here are, as far as I understand, purely based on the variability of the respective lifetimes over roughly a decade. This should be made clear in the caption. Given that the manuscript already contains a comparison with observational data, the derivation of actual lifetime uncertainties (e.g., based on the observed differences) would be highly desirable. As this would undoubtedly create a lot of work, an alternative way forward might be the honest discussion of the probable scale of such uncertainties, so as not to mislead the readership.

*Estimating uncertainties of the loss rates in the model is difficult and there is no clearly defined method to do so. Instead we have split up the lifetimes of SF6 into a mean of the maxima and a mean of the minima to highlight the large amplitude of the seasonal variability in the case of ERA5 and have additionally computed the multi-reanalysis mean (MRM). We computed the root of the squared differences to the MRM as an additional measure of uncertainty due to the choice of reanalysis.*

l285-287 What uncertainty in the lifetime would a +-10% difference in the volume mixing ratio translate into? Also, given the differences between the two satellite data sets, it is not clear whether the word "bias" is appropriate here (as it assumes that at least one of the observational data sets is correct, whereas in reality it could be neither).

*We have changed all occurrences of the word "bias" to "difference", as this is indeed an ambiguous term in this case. Systematic uncertainties of the volume mixing ratios themselves cancel out in the computation of the lifetime. The loss rates and the electron density are the biggest sources of uncertainty. We have done a sensitivity test to demonstrate this. The following sentences were added: "Additionally, a one day simulation was done to assess the sensitivity to the volume mixing ratios of SF6 on the lifetime. Given that BASCOE slightly underestimates the SF6 concentrations, we increased the initial conditions of SF6 by 10% and ran the model for one day to analyse the impact of the volume mixing ratios. We found that the instantaneous lifetime of one snapshot of the model decreased by about 30 years. Similarly, if the volume mixing ratios were decreased by 10%, the lifetime increased by about 40 years. This difference is small compared to the total lifetime, on the order of 1%, which indicates that the chemistry of SF6 is more or less linear, i.e. not highly coupled with other gases. This implies that the lifetime computation is relatively independent of potential errors in the volume mixing ratios."*

l290-293 This is not a balanced assessment of the state of the literature. Several of the lifetime estimates shown in Figure 9 do not agree with the results of this study. This should be mentioned and the potential reasons discussed, including the proper uncertainties of the lifetimes derived here.

*We have added the following sentence: "Our lifetime of CH4 is remarkably higher than that of Volk et al. (1997), which could be due to uncertainties in the OH concentrations. We also find a lower HCFC-22 lifetime than Moore and Remedios (2008), which may similarly reflect differences in modelled OH concentrations."*

l292-293 All other trace gases for which lifetimes were derived here, are much shorter-lived than SF6 and have their dominant sink regions far below the mesosphere. Their ability to confirm the validity of the SF6 lifetime derivation is therefore severely limited, and this should be made clear.

*The computation of the atmospheric lifetime is independent of the absolute lifetime of the species and the method is not affected by the location of the sinks as long as they are within the model domain where chemistry is computed.*

*The following sentence was added in the results section: "While these species have much shorter lifetimes than SF6 and have sinks in different regions, the diagnostic used here is independent of the absolute lifetime for species with well-defined loss processes. The limitation of the validation of the lifetime computation is that losses outside of the model domain are not taken into account. However, BASCOE reaches altitudes of around 80 km where the maximum loss of SF6 occurs (Kouznetsov et al., 2020)."*